# Joint trajectories of physical activity, health, and income before and after statutory retirement: A 22-year follow-up

Tea Lallukka[1]*, Petteri Kolmonen[1,2], Ossi Rahkonen[1], Eero Lahelma[1], Jouni Lahti[1,3]

1 Department of Public Health, University of Helsinki, Helsinki, Finland, 2 LUT School of Engineering Sciences, LUT University, Lappeenranta, Finland, 3 Department of Public Health and Welfare, Finnish Institute for Health and Welfare (THL), Helsinki, Finland

* tea.lallukka@helsinki.fi

**Data Availability Statement:** Data cannot be shared publicly because of their highly sensitive nature, even after pseudonymization. We are not permitted by law to share these data (General Data

## Abstract

### Background

Health behaviors, health, and income change during aging. However, no previous studies have examined, how they develop together over the transition to statutory retirement. We aimed to examine their joint development and to identify the determinants of any distinct trajectories.

### Methods

We studied former employees of the City of Helsinki, Finland, who transitioned to full statutory retirement between 2000 and 2022 (n = 5209, 80% women). We examined five repeated questionnaire surveys to identify any joint developmental patterns in the key indicators of healthy aging and well-being—leisure-time physical activity, health measured by general health perceptions, and household income, over a follow-up of 22 years. We used joint group-based trajectory analysis to identify latent developmental groups. The social and health-related determinants of trajectory group membership are reported as average marginal effects.

### Results

We found four distinct joint trajectory groups. Group 1 (22.6%) had consistently poor general health perceptions, less physical activity than the recommended amount, and low income. In Group 2 (34.2%), general health perceptions were first good but then declined, and income was low but slightly increasing. Group 3 (12.3%) had good general health perceptions, a very high level of physical activity, but fluctuating income. In Group 4 (30.9%), general health perceptions were first good but then declined, physical activity was at the recommended level, and income was sharply increasing. People with obesity had a 22 percentage-point (21–24) higher predicted probability of belonging to Group 1 than people with normal weight. They were also more likely to report low education and more physician-diagnosed chronic diseases and mental disorders.

Protection Regulations, GDPR). More details on the Act are available at: https://fra.europa.eu/en/law-reference/personal-data-act-5231999 (Personal Data Act (523/1999)). These data can only be used by the research group, on a secure server, based on strict confidentiality. This has also been informed to all study participants. The University of Helsinki Information Security Manager Kenneth Kahri (kenneth.kahri@helsinki.fi) has assessed the data sensitivity and based on the data, the decision was that these data can only be stored at a university server, confirmed to be sufficiently secure to manage any risks. Our data protection document available at our webpage includes all the details: https://www.helsinki.fi/en/researchgroups/helsinki-health-study/data-protection-statement. The research ethics committee of the Faculty of Medicine has given a positive statement about the study. Its contact details are found at: https://www.helsinki.fi/en/research/services-researchers/ethical-review-research/medical-research Further details about legal aspects can be asked from the university lawyers tutkimuksenjuristit@helsinki.fi and about the data management from the data support team: datasupport@helsinki.fi. All requests about the original data can be sent to: kttl-hhs@helsinki.fi. The emails to the address are accessed only by the members of the Helsinki Health Study group and the server is maintained by the Institution, the University of Helsinki. Inquiries can also be sent to the PI of the project, Professor Tea Lallukka (tea.lallukka@helsinki.fi), who also has access to the project email kttl-hhs@helsinki.fi.

**Funding:** This study was supported by the Research Council of Finland (Grant #330527). OR was supported by the Ministry of Education and Culture, Finland (Grant #OKM/76/626/2023) and by the Juho Vainio Foundation (Grant #202300041). The funders had no role in study design, data collection and analysis, decision to publish, or preparation of the manuscript.

**Competing interests:** The authors have declared that no competing interests exist.

## Conclusions

We identified distinct trajectories in physical activity, general health perceptions, and income over a follow-up of over 20 years. The majority of those who had transitioned to statutory retirement had good general health perceptions but varying levels of physical activity and income. As not all those with a low income had a low level of physical activity or poor general health perceptions, public health interventions should target distinct groups with the most adverse risk factor profiles, to narrow health inequalities during aging.

## Introduction

Health behaviors, health and socioeconomic position, particularly income are known to change as a person ages [1–4]. Although changes in health belong to a normal biological process, the speed and patterns of change vary among different individuals and population subgroups. Moreover, life expectancy is steadily increasing [5], and it has been projected that the share of people aged 65 or more will roughly double by 2050 in Europe alone [6]. This highlights the need to understand how health and its determinants develop in the aging population, and to identify what drives healthy aging. Age itself is unlikely to be a sufficient marker [7]. Different underlying, potentially preventable factors may be associated with unfavorable changes. Other factors over the life course could act as buffers and help explain why some people maintain physically active lifestyles and good health.

Numerous studies have addressed physical activity [8–11]. Some have found positive associations but have also faced criticism regarding potential publication bias and low quality [9]. High levels of physical activity are mostly consistently associated with better health and functioning whereas physical inactivity is associated with worse health and disability [11]. Furthermore, inequalities in health behaviors and health are substantial and persistent, and the gap in life expectancy between people with the highest and lowest incomes has even widened in Finland [12]. However, none of the above previous variable-oriented studies have considered latent groups or examined how well-being indicators develop together, for example, changes over a major life transition, statutory retirement, or following the same people from their mid- or later careers until older age. To promote health, prevent health risks, and tackle health inequalities, we need to better understand the development of risk factors, health, and inequalities, and the factors that might explain distinct trajectories. The importance of this is highlighted in contemporary societies, in which populations are aging, and the need for health care is increasing due to the population's declining health [13, 14]. Novel information on the distinct development of health and well-being could help identify new risk groups for more timely prevention and health promotion.

Following the joint development of physical activity, health, and income can help confirm whether they co-occur. In our model, we let the data speak and identified the latent groups that approximated true development in different outcomes over the follow-up. We looked at, for example, whether some groups were continuously better off in each indicator, or whether the development of each indicator was distinct. If physical inactivity, poor health, and low income co-occur over time, early interventions are needed already long before retirement, to promote healthier aging. The purpose of the model was not to confirm causal associations but to identify distinct developmental patterns, placing each individual in a group that was most likely to describe their physical activity, income, and health across specific time points.

Moreover, as several sociodemographic and socioeconomic factors, such as physical working conditions, sleep and obesity, have been linked to physical activity, health and income [15–18], they may potentially be linked to distinct developmental patterns in these outcomes.

We aimed to examine the joint development of leisure-time physical activity, general health perceptions, and household income among aging women and men, before and after their transition to statutory retirement, using the person-oriented method of joint Group-based Trajectory Modeling (joint GBTM) [19–21]. This method enabled us to approximate whether or not these indicators of health and well-being co-occur. A further aim was to identify the social and health-related determinants of distinct developmental patterns.

## Materials and methods

### Data

The data for the study were retrieved from the ongoing Finnish Helsinki Health Study (HSS), which is a follow up study of a cohort of initially employed women and men, who have responded to repeated mailed questionnaires before and after their retirement [22]. The Phase 1 data were collected in 2000, 2001 and 2002, when the target population were 40- to 60-year-old employees of the City of Helsinki, Finland. The City of Helsinki is the largest employer in Finland, with around 38 000 employees. Most of the midlife and older employees studied had a permanent contract and continued working until their retirement. This makes the cohort ideal to study health and its development, and explanatory factors before and after retirement transition. Altogether 8960 employees responded to the baseline survey (response rate 67% of the target population). Follow-up surveys were mailed to all Phase 1 respondents in 2007 (Phase 2), 2012 (Phase 3), 2017 (Phase 4) and 2022 (Phase 5), irrespective of their employment status at each phase. No new respondents were added at any time point. The response rates to the follow-up phases were 83%, 79%, 82% and 75%, respectively. All the outcomes included in this study were repeated at each phase, enabling their joint development to be modeled over the 22-year follow-up. The first response for Phase1 was received on April 6, 2000, and the last for Phase 5 on September 28, 2022. The participants received all the required information in the cover letter and were also mailed an additional leaflet describing the study. The project website contains all the information, and makes it publicly available, including the data protection statement with details on giving consent and all the legalities concerning personal data processing (helsinki.fi/hhs). The participants further provided their signed informed consent to their survey responses being linked to register data, such as occupational class from the employer's register.

For the purposes of this study, we included all former employees of the City of Helsinki who retired during the study follow-up between 2000 and 2022, i.e., after Phase 1 and before Phase 5. During the follow-up, about 90% retired. In each follow-up survey, they reported whether they had retired, when they had retired, and the type of their retirement (please see S1 File for the survey items in 2022, Question 2a, b and c). From the information, we extracted those who transitioned to full statutory retirement and computed the retirement years. Those on part-time retirement were considered as being in employment and removed from the data as were all other than full-time statutory retirees. The average follow-up time was 18.7 years. A total of 79.4% were followed for 20 years, 15.9% for 15 years, and 4.7% for a minimum of 10 years (3 time points). As the participants could retire at any time after Phase 1, their follow-up times varied before and after their transition into retirement. This was considered in the model and the outcomes were modeled on the retirement year, not the phase. Of the participants, 80% were women, corresponding to the target population and the Finnish public sector in general [22].

Before 2005, old-age retirement age was 65 in Finland, but from 2005 onwards and throughout the study period, retirement age has been flexible from 63 to 68 years. During the study period, public-sector employees could also have personal retirement ages, typically ranging from 63 to 65. Some also had a fixed occupational retirement age, which could have been lower than 63. This is why we considered both age and retirement age in our models. More details of the system are available in our earlier study and in the Statistical Yearbook of Pensioners in Finland [3, 23].

We used data from five repeated surveys to examine the joint developmental patterns in the indicators of well-being, leisure-time physical activity, household income, and health. Next, we studied the associations between social and health-related factors and distinct developmental patterns. All those who had responded to at least two of the five follow-up questionnaires were included in our study (n = 5209). As in all surveys, all the data were self-reported.

This study was approved by and received research permission from the health authorities of the City of Helsinki. The study plan was approved by the Ethical Committee of the Faculty of Medicine, University of Helsinki, Finland.

## Outcomes

**Leisure-time physical activity.** Four survey items elicited the frequency and intensity of leisure-time (including commuting) physical activities. As commuting was part of the items, it cannot be studied separately. The items repeated at each time point asked how for many hours the participants had engaged in activities equivalent to walking, brisk walking, jogging, and running over the past 12 months. The responses were transformed into metabolic equivalent values (METs), following previous procedures and guidelines [24, 25]. Walking was given a weight of 4, brisk walking a weight of 6, jogging a weight of 10, and running or similar activities a weight of 13. If the participant reported spending no time on an activity, it was coded as 0 minutes. The analyses included all those who had responded to at least one of the four items. Missing responses (0.7–1.3% per phase) were coded as 0 minutes for all those who had responded to at least one of the four questions. All MET data was missing from at least one phase for 4.4%, and these were excluded. Estimated MET-hours per week comprise information on the hours spent on leisure-time physical activities and their intensity, and have been employed in numerous previous studies using these same data, and elsewhere [1, 26, 27].

**General health perceptions.** General health perceptions were measured using the general health perceptions subscale from the widely-used RAND-36 inventory [28, 29]. This subscale comprises five items aiming to capture respondents' opinions on their current self-reported health, susceptibility to diseases, and health development [30]. The respondents were asked to rate their health according to five response alternatives ranging from poor to excellent. The score, ranging from 0–100, represents perceived health, from excellent to poor and declining. The first item of the subscale, called self-rated health, is a powerful measure, which has been used for decades as a single item to predict disability retirement [31], morbidity, and mortality [32]. It has also shown good reliability across different population subgroups [33]. In this study, we preferred to use a wider subscale with more information on health perceptions and health development. As there were basically no missing data in the score, those with missing data on the subscale were excluded from the analyses (1–5 participants, 0%).

**Household income.** At each time point, the respondents estimated the level of their net household income after taxes in an average month. This included all income from paid employment and any welfare benefits or other sources of income such as pension for all members of the household. The reported household income was weighted by the number of children and adults living in it (Respondent 1, other adults 0.5, children younger than 18 0.3), in

line with recommended procedures [34, 35]. We chose household income because most participants lived with other people and thus it provided a more accurate picture of their financial situation and any changes in it, for instance, marital status changing during the follow-up, or children moving out to live on their own, or when any debt or mortgage was fully paid, which affected disposable income. Household income was missing for 2% of participants. These participants were excluded from the analytical sample.

**Social and health-related covariates.** We included a range of social and health-related factors as potential determinants of trajectory group membership and distinct development in physical activity, general health perceptions and household income [15–17]. First, gender was dichotomized as women and men. Marital status was dichotomized to distinguish between participants with a partner (married and cohabiting), and those without (single, divorced, or widowed). Education was categorized into four groups to distinguish between higher education, upper secondary education, lower secondary education, and low-level education [Please see Methods S1 File for more details]. We also adjusted for physically strenuous work before the transition to retirement, and age during the first study wave. As health-related factors, we included sleep duration, body mass index (BMI), smoking, binge drinking, and number of physician-diagnosed chronic diseases and mental disorders. Following the updated recommendations for older adults [36], average sleep was classified as seven to eight hours of sleep, compared to shorter sleep (less than 7 hours), and longer sleep (more than 8 hours [36]. This cut-off point was also chosen to distinguish long sleep from recommended sleep (3.5% slept 9 hours but only 0.2% reported sleeping 10 hours or more). Sleep hours were requested in full hours. BMI was computed as weight (kg) divided by height squared ($m^2$). As less than 1% of participants had underweight (BMI<18.5), they were included in the group of people with normal weight. As BMI was only a covariate and the proportion of individuals with underweight was very small, this was the best procedure to avoid losing participants and causing selection in the analytical data. Smoking was divided into never smoking, ex-smoking, and current smoking. Binge drinking was defined as drinking more than six units of alcohol (beer, wine or spirits) on a single occasion once a week or more often. It was chosen as a covariate, as it is a clear and undisputable risk factor for loss of disease-free life years for example, as shown in our previous multicohort and biobank study, pooling data from 129 942 adults across 12 cohort studies (1.73 million person-years at risk) [18]. Thus, binge drinking could most effectively capture the potential harmful contributions of drinking to the development of our outcomes. Any physician-diagnosed, typically chronic somatic diseases, such as different cardiovascular diseases, diabetes, musculoskeletal diseases, or migraine; and mental health disorders including anxiety, depression, and other mental disorders were summed to indicate prior physician-diagnosed health problems. The summed variable had four categories: 0 diseases, 1, 2, or 3 or more physician-diagnosed chronic diseases and mental disorders. This variable was included to control for the baseline situation, prior to examining changes in other health-related outcomes.

There were no missing values for gender or age. As marital status and education also had a very low number of missing values (n = 39, 1% and n = 29, 1%, respectively), they were not imputed or replaced. Physical work also had overall few missing values (1%). For obvious reasons, after retirement most respondents did not fill in the items concerning working conditions, as they were no longer applicable. Thus, last known physical work was considered. Health-related variables generally only had a few missing values, ranging from 1% to 2% per variable. However, for physician-diagnosed diseases, around 10–14% of the responses were missing. This may be because the respondent did not have the disease in question, and the sum of diseases was computed from the positive responses to having a disease. For sleep duration and BMI, missing values (2% and 1%) were replaced by mean values. Otherwise, we

included all missing values in the reference category in line with previous procedures [37]. This strategy was also to avoid exaggeration of any associations and selective loss of data.

All the survey items used in this study are provided in a supplement with further details on the formulations of the questions and their response alternatives.

## Statistical methods

Joint GBTM was utilized in this study as it can be used to identify latent groups of individuals with distinct developmental trajectories in the measured variables [20, 38]. The strength of GBTM lies in its ability to detect distinct clusters of trajectories within studied populations even when clear-cut classifications for different developmental trajectories cannot otherwise be made [39]. These clusters are formed without reliance on any *ad hoc* assumptions or subjective evaluations, and the method utilizes the data to form approximations of trajectory groups. Joint trajectory modeling assumes that the measured variables do not develop independently from one another and that their trajectories follow similar patterns due to the co-occurrence of these variables [40]. Its advantage is that it can model the development of different outcomes at the same time: in this study, physical activity, general health perceptions and income.

A key component of GBTM is choosing how many trajectory groups to have in the model. Choosing a model with a certain number of trajectory groups should prioritize the inclusion of as many groups as are needed to summarize the data into distinct groups in a useful manner. If adding a new trajectory group does not provide new information to the model, and it is not distinguishable from other trajectory groups in terms of its characteristics or outcomes, its inclusion in the model cannot be justified. In this study, the optimal number of trajectory groups was four. The optimal number of trajectory groups can be determined using multiple methods, including the Bayesian Information Criterion (BIC) [20]. However, as the BIC values for the model did not converge to a clear minimum point (see S1 Table in S1 File) for the tested number of trajectory groups ranging from 1 to 10, the optimal number of groups was settled on as four, as any larger number of groups resulted in them no longer being adequately distinct from one another.

Although it is not possible to determine group membership for everyone in definitive terms, it can be probabilistically estimated using posterior probability [39]. Once the model has been established, the posterior probabilities for group membership are calculated for all individuals in the study population. These individuals are then assigned to groups on the basis of their highest posterior probability. Each person belongs to the group that most closely corresponded to their individual qualities and development in the well-being outcomes. The mean posterior probabilities can be calculated for each group as a measurement of the model's fit and reliability. S2 Table in S1 File presents the mean posterior probabilities for the four trajectory groups. The means range from 0.90 to 0.93, which provides adequate support for the model's goodness of fit. Although GBTM is useful for identifying latent groups likely to follow similar developments in health or social outcomes such as ours, it may also produce spurious trajectories if the criteria for evaluating and selecting the best model are not adequately considered [41]. Thus, we conducted a series of different model fits and other tests to confirm our selected model. The results used in the model selection are available as Supplementary material (S1 and S2 Tables and S1 and S2 Figs in S1 File). Although some statistical criteria could point to more complex five- or six-class solutions, we decided that a comprehensive assessment that considers both statistical criteria and meaningful interpretation, a four-group model, was best (S3 Fig in S1 File). This is because the more complex models (5–6 classes or more) provided no new information, as some classes did not remain clearly distinct. They were also too small for further analytical steps, such as studying associations with social and health-related factors, which require models with fewer and larger classes.

We then examined the sociodemographic composition of the selected four trajectory groups and cross-tabulated the distributions for social and health-related determinants with the trajectory group. Finally, the associations between social and health-related determinants and trajectory group membership were analyzed using multinomial logistic regression models, with a margins option to produce average marginal effects (AME). Models produce AMEs and their 95% confidence intervals, which is a a more concrete option for comparing different groups. A key advantage of AMEs is that the reference group trajectory also has estimates, whereas for odds ratios, the reference trajectory is given a value of 1 and other trajectory groups are compared to this. AMEs can be considered more concrete, and the group sizes are unlikely to cause bias or exaggerate the estimates, which is common with odds ratios. Odds ratios have also been criticized earlier, and their differences from AMEs have been discussed [42]. The reported AME values show the predicted probabilities of membership in the four different trajectory groups in relation to the reference category of each social and health-related determinant. Despite the criticism, as well as for comparison, we also report the odds ratios as a supplementary analysis (please see S3 Table in S1 File). All the analyses in this study were conducted using the R 4.2.1 software. We used the FlexMix package for forming the trajectory group model [43] and the marginal effects package for calculating the AMEs.

## Results

### Descriptive results

Table 1 presents the participants' characteristics in Phase 1 (mean age of retirement in S4 Table in S1 File). The majority (70.7%) were married or cohabiting, and around 28.0% had

**Table 1. Distributions of sociodemographic and health variables in the Helsinki Health Study population during Phase 1 in 2000–2002.**

| | Study population [n = 5209] | | Study population [n = 5209] |
|---|---|---|---|
| Gender | | Smoking | |
| Men | 1019 (19.6%) | No smoking | 2815 (54.0%) |
| Women | 4190 (80.4%) | Past smoking | 1049 (20.1%) |
| | | Current smoking | 1312 (25.2%) |
| Marital status | | Binge drinking | |
| Cohabiting/married | 3684 (70.7%) | No | 4601 (88.3%) |
| Never married/divorced/widowed | 1496 (28.7%) | Binge drinking (once a week or more) | 491 (9.4%) |
| Education | | Number of physician-diagnosed chronic diseases and | |
| Higher education | 1456 (28.0%) | mental disorders | |
| Upper secondary | 1526 (29.3%) | 0 | 2517 (48.3%) |
| Lower secondary | 1075 (20.6%) | 1 | 1615 (31.0%) |
| Low-level | 1113 (21.4%) | 2 | 687 (13.2%) |
| | | 3 or more | 390 (7.5%) |
| Average hours of sleep | | Physically strenuous work | |
| 7–8 hours | 3771 (72.4%) | No | 1115 (21.4%) |
| Less than 7 hours | 1286 (24.7%) | Low level of physically strenuous work | 2283 (43.8%) |
| 9 hours or more | 182 (3.5%) | High level of physically strenuous work | 1735 (33.3%) |
| Obesity | | Age | |
| Normal weight | 2566 (49.3%) | 54 or younger | 2591 (49.7%) |
| Overweight | 1891 (36.3%) | 55–59 | 1768 (33.9%) |
| Obesity | 752 (14.4%) | 60 or older | 850 (16.3%) |

Due to missing answers (on average 1–2% per item), some of the distributions presented in the table do not add up to 100%

higher education and about a fifth (21%) had low-level education. A total of 72.4% of the participants reported sleeping seven to eight hours, and 24.7% and 3.5% reported sleeping less and more, respectively. Around half of the participants had a healthy weight, 36.3% were with overweight, and 14.4% with obesity. About a fourth (25.2%) of the participants were current smokers, a fifth (20.1%) were ex-smokers, and 9.4% reported binge drinking. Physically strenuous work was common before retirement, as about a third of the participants reported having done physically highly strenuous work, and 43.8% reported a low level of physically strenuous work. Only 21.4% had no history of physically strenuous work.

## Joint development of leisure-time physical activity, general health perceptions, and household income

We found four distinct groups describing the joint trajectories of leisure-time physical activity, general health perceptions, and household income over the follow-up of up to 22 years, including up to 20 years before and after statutory retirement transition (Fig 1). Those in Group 1 (22.6%) had consistently poor general health perceptions and clearly less than the recommended amount of leisure-time physical activity (14.0–16.6 MET-hours/ week through phases 1–5, corresponding to about *2.5 hours of brisk walking*), and low, although increasing income, particularly before retirement. Those in Group 2 (34.2%) had first had good general health perceptions, but this had declined, and had the recommended amount of leisure-time physical activity (28.8–30.6 MET-hours /week through phases 1–5, corresponding to about *5 hours of brisk walking)*, and low but increasing income, particularly before retirement, as in Group 1. Those belonging to Group 3 (12.3%) had good general health perceptions, and a very high, albeit later a slightly decreasing level of leisure-time physical activity (60.4–63.3 MET-hours/ week through Phases 1–5, corresponding to about *5 hours of brisk walking and 2.5 hours of running*), and an average and fluctuating income (first increasing, then decreasing). Despite the decrease toward the end of the follow-up, the level of physical activity remained high and was the highest of all the groups throughout the follow-up. Finally, those in Group 4 (30.9%) had initially good but then declining health as measured by general health perceptions. They also had the recommended, although slightly lower levels of leisure-time physical activity than those in Group 2 (23.4–26.9 MET-hours /week through Phases 1–5, corresponding to about *4 hours of brisk walking*), and sharply increasing and then high income. Overall, these results and developmental patterns highlight that in most of the groups, the respondents had good general health perceptions both before and after their retirement, mostly irrespective of developments in household income. Thus, only one group had continuously poor general health perceptions, low levels of physical activity, and low, although increasing income levels, and comprised less than a fourth of the participants. Likewise, only one group had a continuously very high level of physical activity and good general health perceptions but varying income.

## Associations between social and health-related factors and trajectory group membership

After identifying the trajectory groups, we first descriptively examined how the included social and health-related factors were associated with trajectory group membership (Table 2). The results show clear differences between the trajectory groups, in particular in terms of gender composition, marital status, education level, obesity, smoking, number of physician-diagnosed chronic diseases and mental disorders, and history of physically strenuous work.

Of all the studied social and health-related determinants of trajectory membership, education and marital status stood out as the social determinants, and obesity and physician-diagnosed chronic diseases and mental disorders as the health-related determinants of group

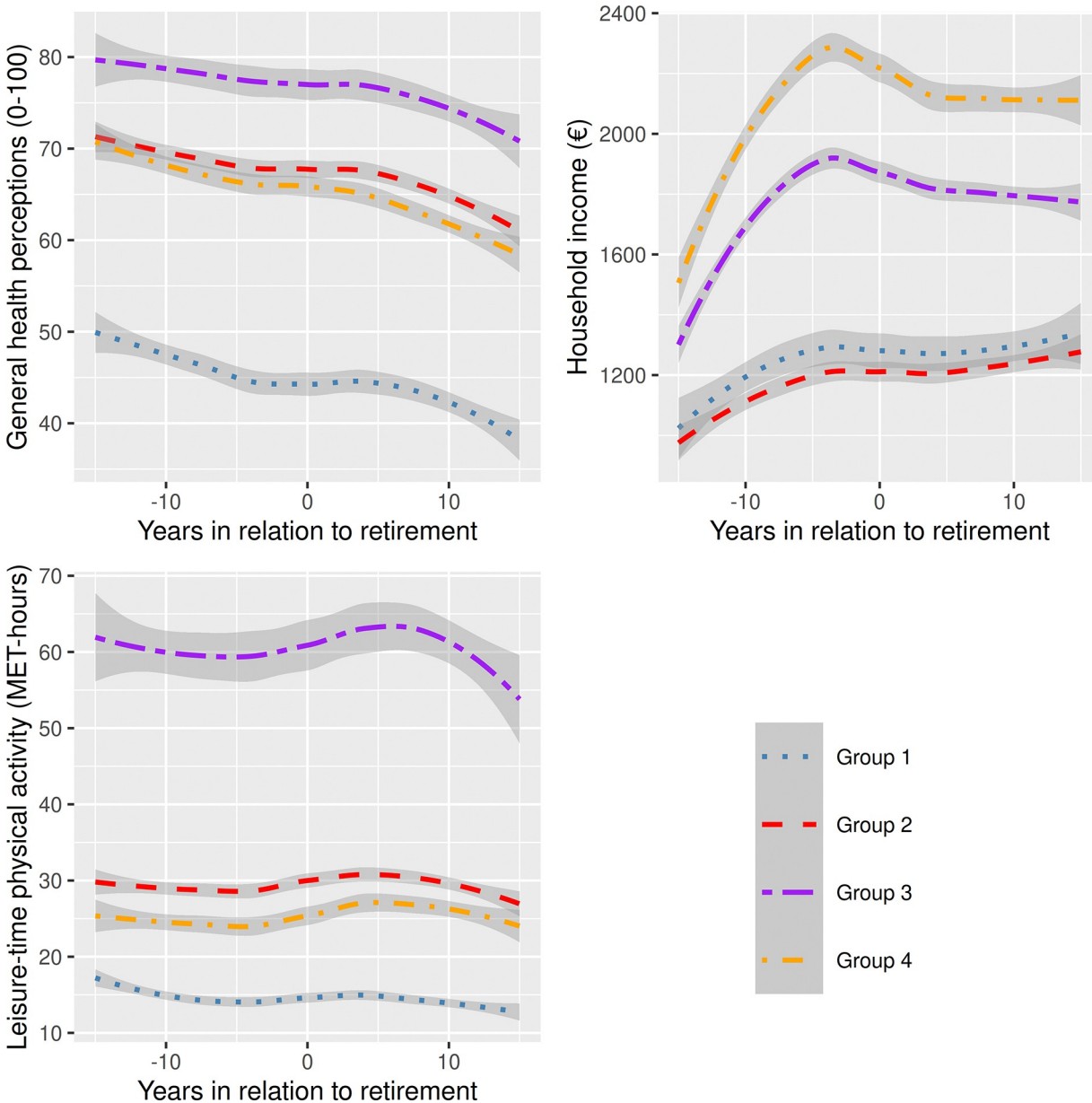

**Fig 1. Joint development of leisure-time physical activity (metabolic equivalent, MET, hours), general health perceptions (score 0–100), and household income (€) 10 years before and after statutory retirement (x-axis) among the Helsinki Health Study participants 2000–2022 (n = 5209).** Joint Group-based Trajectory Modeling: Group 1 (22.6%), Group 2 (34.2%), Group 3 (12.3%), Group 4 (30.9%).

membership (Table 3). Thus, those with obesity had a 22 percentage-point higher predicted probability of belonging to Group 1 than those with healthy weight. Reporting three or more physician-diagnosed chronic diseases and mental disorders was associated with a 24 percentage-point higher predicted probability of belonging to Group 1, and 11 and 10 percentage-point lower predicted probability of belonging to Group 2 and 3, respectively. However, physician-diagnosed chronic diseases and mental disorders were not associated with trajectory membership of Group 4. Group 4 was highly educated, as was Group 3, while those with low-level education were less likely to be assigned to Group 4. Instead, people with low education were more likely to belong to trajectory Group 1, particularly Group 2. Those who were

**Table 2. Distributions[a] of social and health-related variables in trajectory groups during Phase 1 in 2000–2002.**

| | Study population [n = 5209] | Group 1 [n = 1177] | Group 2 [n = 1783] | Group 3 [n = 639] | Group 4 [n = 1610] |
|---|---|---|---|---|---|
| Gender | | | | | |
| Men | 1019 (19.6%) | 198 (16.8%) | 232 (13.0%) | 199 (31.1%) | 390 (24.2%) |
| Women | 4190 (80.4%) | 979 (83.1%) | 1551 (87.0%) | 440 (68.9%) | 1220 (75.8%) |
| Marital status | | | | | |
| Cohabiting/married | 3684 (70.7%) | 738 (62.7%) | 1040 (58.4%) | 504 (78.9%) | 1402 (87.1%) |
| Never married/divorced/widowed | 1496 (28.7%) | 431 (36.6%) | 735 (41.2%) | 132 (20.7%) | 198 (12.3%) |
| Education | | | | | |
| Higher education | 1456 (28.0%) | 188 (16.0%) | 261 (14.6%) | 227 (35.5%) | 780 (48.4%) |
| Upper secondary | 1526 (29.3%) | 323 (27.4%) | 487 (27.3%) | 204 (31.9%) | 512 (31.8%) |
| Lower secondary | 1075 (20.6%) | 308 (26.2%) | 501 (28.1%) | 104 (16.3%) | 162 (10.1%) |
| Low-level | 1113 (21.4%) | 347 (29.5%) | 520 (29.2%) | 101 (15.8%) | 145 (9.0%) |
| Average hours of sleep | | | | | |
| 7–8 hours | 3771 (72.4%) | 788 (66.9%) | 1281 (71.8%) | 479 (75.0%) | 1193 (74.1%) |
| Less than 7 hours | 1286 (24.7%) | 353 (30.0%) | 432 (24.2%) | 140 (21.9%) | 361 (22.4%) |
| 9 hours or more | 182 (3.5%) | 36 (3.1%) | 70 (3.9%) | 20 (3.1%) | 56 (3.5%) |
| Obesity | | | | | |
| Normal weight | 2566 (49.3%) | 352 (29.9%) | 928 (52.0%) | 453 (70.9%) | 833 (51.7%) |
| Overweight | 1891 (36.3%) | 487 (41.4%) | 662 (37.1%) | 167 (26.1%) | 575 (35.7%) |
| Obesity | 752 (14.4%) | 338 (28.7%) | 193 (10.8%) | 19 (3.0%) | 202 (12.5%) |
| Smoking | | | | | |
| No smoking | 2815 (54.0%) | 580 (49.3%) | 972 (54.5%) | 364 (57.0%) | 899 (55.8%) |
| Current smoking | 1049 (20.1%) | 320 (27.2%) | 394 (22.1%) | 89 (13.9%) | 246 (15.3%) |
| Past smoking | 1312 (25.2%) | 270 (22.9%) | 405 (22.7%) | 185 (29.0%) | 452 (28.1%) |
| Binge drinking | | | | | |
| No | 4601 (88.3%) | 1009 (85.7%) | 1621 (90.9%) | 574 (89.8%) | 1397 (86.8%) |
| Binge drinking (once a week or more) | 491 (9.4%) | 136 (11.6%) | 120 (6.7%) | 53 (8.3%) | 182 (11.3%) |
| Number of physician-diagnosed chronic diseases and mental disorders | | | | | |
| 0 | 2517 (48.3%) | 360 (30.6%) | 903 (50.6%) | 415 (64.9%) | 839 (52.1%) |
| 1 | 1615 (31.0%) | 403 (34.2%) | 538 (30.2%) | 155 (24.3%) | 519 (32.2%) |
| 2 | 687 (13.2%) | 221 (18.8%) | 227 (12.7%) | 53 (8.3%) | 186 (11.6%) |
| 3 or more | 390 (7.5%) | 193 (16.4%) | 115 (6.4%) | 16 (2.5%) | 66 (4.1%) |
| Physically strenuous work | | | | | |
| No | 1115 (21.4%) | 173 (14.7%) | 270 (15.1%) | 188 (29.4%) | 492 (30.6%) |
| Low level of physically strenuous work | 2283 (43.8%) | 470 (39.9%) | 730 (40.9%) | 290 (45.4%) | 793 (49.3%) |
| High level of physically strenuous work | 1735 (33.3%) | 514 (43.7%) | 758 (42.5%) | 166 (26.0%) | 297 (18.4%) |
| Age | | | | | |
| 54 or younger | 2591 (49.7%) | 502 (42.7%) | 827 (46.4%) | 392 (61.3%) | 870 (54.0%) |
| 54–59 | 1768 (33.9%) | 425 (36.1%) | 641 (36.0%) | 188 (29.4%) | 514 (31.9%) |
| 60 or older | 850 (16.3%) | 250 (21.2%) | 315 (17.7%) | 59 (9.2%) | 226 (14.0%) |

[a]Due to missing answers (on average <2% per item), some of the distributions presented in the table do not add up to 100%.

neither married nor cohabiting had a 22 percentage-point higher predicted probability of belonging to Group 2, and 24 percentage-point lower predicted probability of belonging to Group 4. In our additional analyses with odds ratios, mostly the same factors emerged as determinants of group membership (S3 Table in S1 File). Many of the odds ratios were high, but as

**Table 3. Average Marginal Effects (AME) for trajectory groups in a multinomial logistic regression (with 95% confidence intervals) [Age and retirement age included].**

| | Group 1 [n = 1177] | Group 2 [n = 1783] | Group 3 [n = 639] | Group 4 [n = 1610] |
|---|---|---|---|---|
| Gender | | | | |
| Men | - | - | - | - |
| Women | -0.02 (-0.04 \| -0.01) | 0.09 (0.07 \| 0.10) | -0.07 (-0.09 \| -0.06) | -0.02 (-0.03 \| -0.02) |
| Marital status | | | | |
| Cohabiting/married | - | - | - | - |
| Never married/divorced/widowed | 0.05 (0.04 \| 0.07) | 0.22 (0.21 \| 0.23) | -0.03 (-0.04 \| -0.02) | -0.24 (-0.26 \| -0.23) |
| Education | | | | |
| Higher education | - | - | - | - |
| Upper secondary | 0.04 (0.03 \| 0.06) | 0.13 (0.11 \| 0.14) | 0.00 (-0.01 \| 0.01) | -0.17 (-0.18 \| -0.15) |
| Lower secondary | 0.08 (0.06 \| 0.09) | 0.25 (0.23 \| 0.27) | -0.02 (-0.03 \| 0.00) | -0.31 (-0.33 \| -0.30) |
| Low-level | 0.12 (0.10 \| 0.13) | 0.25 (0.24 \| 0.27) | -0.03 (-0.04 \| -0.02) | -0.34 (-0.36 \| -0.33) |
| Average hours of sleep | | | | |
| 7–8 hours | - | - | - | - |
| Less than 7 hours | 0.05 (0.04 \| 0.06) | -0.01 (-0.02 \| 0.00) | -0.02 (-0.03 \| -0.01) | -0.03 (-0.04 \| -0.01) |
| 9 hours or more | 0.01 (-0.01 \| 0.03) | 0.01 (-0.01 \| 0.04) | -0.01 (-0.03 \| 0.01) | -0.02 (-0.04 \| 0.01) |
| Obesity | | | | |
| Normal weight | - | - | - | - |
| Overweight | 0.08 (0.07 \| 0.09) | -0.02 (-0.03 \| 0.00) | -0.09 (-0.10 \| -0.08) | 0.03 (0.02 \| 0.04) |
| Obesity | 0.22 (0.21 \| 0.24) | -0.10 (-0.12 \| -0.09) | -0.15 (-0.16 \| -0.14) | 0.02 (0.01 \| 0.04) |
| Smoking | | | | |
| No smoking | - | - | - | - |
| Current smoking | 0.09 (0.07 \| 0.10) | 0.00 (-0.02 \| 0.02) | -0.05 (-0.06 \| -0.04) | -0.04 (-0.05 \| -0.02) |
| Past smoking | -0.02 (-0.03 \| -0.01) | -0.01 (-0.02 \| 0.00) | 0.02 (0.01 \| 0.03) | 0.02 (0.00 \| 0.03) |
| Binge drinking | | | | |
| No | - | - | - | - |
| Binge drinking (once a week or more) | 0.04 (0.02 \| 0.06) | -0.03 (-0.05 \| -0.01) | -0.02 (-0.03 \| 0.00) | 0.01 (-0.01 \| 0.03) |
| Number of physician-diagnosed chronic diseases and mental disorders | | | | |
| 0 | - | - | - | - |
| 1 | 0.05 (0.04 \| 0.06) | -0.03 (-0.05 \| -0.02) | -0.04 (-0.05 \| -0.02) | 0.02 (0.00 \| 0.03) |
| 2 | 0.11 (0.10 \| 0.13) | -0.06 (-0.08 \| -0.04) | -0.06 (-0.07 \| -0.05) | 0.01 (-0.01 \| 0.02) |
| 3 or more | 0.24 (0.23 \| 0.26) | -0.11 (-0.13 \| -0.09) | -0.10 (-0.11 \| -0.09) | -0.03 (-0.05 \| -0.01) |
| Physically strenuous work | | | | |
| No | - | - | - | - |
| Low level of physically strenuous work | 0.03 (0.02 \| 0.05) | 0.05 (0.04 \| 0.07) | -0.02 (-0.04 \| -0.01) | -0.06 (-0.07 \| -0.04) |
| High level of physically strenuous work | 0.08 (0.06 \| 0.09) | 0.08 (0.06 \| 0.10) | -0.02 (-0.04 \| -0.01) | -0.14 (-0.15 \| -0.12) |
| Age | | | | |
| 54 or younger | - | - | - | - |
| 54–59 | -0.02 (-0.04 \| 0.00) | 0.02 (0.00 \| 0.04) | -0.01 (-0.02 \| 0.01) | 0.00 (-0.02 \| 0.02) |
| 60 or older | -0.02 (-0.04 \| -0.01) | 0.03 (0.02 \| 0.05) | -0.01 (-0.03 \| 0.00) | 0.01 (-0.01 \| 0.02) |
| Age of retirement | | | | |
| 59 or younger | - | - | - | - |
| 60–64 | -0.01 (-0.04 \| 0.02) | -0.03 (-0.07 \| -0.01) | -0.03 (-0.06 \| -0.01) | 0.07 (0.04 \| 0.11) |
| 65 or older | -0.01 (-0.04 \| 0.02) | -0.03 (-0.06 \| 0.01) | -0.03 (-0.05 \| 0.01) | 0.06 (0.03 \| 0.10) |

the groups were only compared to the reference trajectory (Group 3), the odds ratios only showed how those in Groups 1, 2 and 4 differed from those in Group 3. People in Groups 1 and 2 were more likely to be women and single; to have lower-level education, a history of

physically strenuous work and smoking, and a high BMI; and to report poorer somatic and mental health than those in Group 3. Those in Group 4 were in turn less likely to have low-level educated or to be single, but were also more likely to be women, have obesity, and report physician-diagnosed chronic diseases and mental disorders. However, the very high odds ratios in these additional analyses should be cautiously interpreted.

## Discussion

### Main findings

This study sought to examine the joint development of leisure-time physical activity, general health perceptions, and household income before and after statutory retirement. The main finding of this study was the distinct joint development in these outcomes among those who had transitioned to statutory retirement. The results highlight that most of these retirees had good health as measured by general health perceptions before and after their retirement, irrespective of the development of their income or physical activity. One group had continuous poor general health perceptions, low levels of physical activity, and low, although increasing, income levels. Another group had very high levels of physical activity, good general health perceptions but low income. We also found that changes in income did not consistently develop jointly with changes in general health perceptions, which adds to the understanding of social inequalities in health, and highlights that the development is heterogeneous and low income is not always associated with poor health. Such developmental patterns and their explanations need to be considered in efforts to narrow health inequalities. Higher levels of physical activity are consistently linked with better health, and doing the recommended amount of physical activity was associated with higher scores in general health perceptions among aging and older adults. Finally, we examined the social and health-related factors that were associated with distinct developmental patterns. Among the potential factors, education and obesity were key factors that explained the patterns. Marital status and physician-diagnosed chronic diseases and mental disorders may also have contributed to the distinct development to some extent, but gender was mainly unassociated with any distinct development in the three jointly studied outcomes. However, the choice of estimates (AMEs vs. odds ratios) may have affected the difference in the strength of the associations.

### Interpretation

Physical activity typically declined with age but was important for general health perceptions [11]. Changes in income in turn did not develop jointly with changes in general health perceptions in all the groups. It is possible that the decline in physical activity in some groups reflect decreasing intensity, as MET-hours may decrease although the duration of the activities might remain the same. Income differences appeared to narrow during the follow-up and after retirement, except in one group, in which it increased sharply and became high. One explanation for this could be that a pension is typically lower than salary from paid employment, and we focused on household income, which can differ from individual income and have different or more complex associations with health-related outcomes [44]. Household income, nonetheless, is likely to better reflect the actual material situation of the respondent, as it takes into account other people living with the respondent and their sources of income. It is difficult, however, to compare any of our findings to those of previous studies, as they are mostly variable oriented and have focused on the associations between physical activity and health [8, 11], or on the inequalities in physical activity and health by income [12, 45, 46]. Instead, our study modeled their joint development over decades and across a major life event, i.e., the transition to statutory retirement. A study comprising evidence from eight cohort studies reported that

physical activity had beneficial effects on health trajectories [47] but did not assess their joint development. Socioeconomic variables were considered covariates only. We identified a group of people with both a relatively low level of physical activity and poor health as measured by general health perceptions, but the results highlight that there is no one single homogeneous group of people with low income and equally poor health development. Therefore, it is important to distinguish between different developmental patterns and their reasons when projecting future health. As a note, these identified latent groups do not clearly support the earlier findings on the curvilinear associations between household income and self-rated health [46], as even sharp increases in income were linked with declining health. This variation could be explained by the use of the general health perceptions subscale [30] instead of the single item derived from the subscale. It may also be due to our design, as we focused on distinct developmental patterns whereas the earlier study used advanced modeling techniques and locally weighted linear regression to smooth the data. Furthermore, as our study further comprised a notably older cohort and more recently collected data, age-period-cohort effects could play a role in the differences between the findings [48]. We also identified large groups of people—two thirds of the current population—who had the recommended and high levels of physical activity, who continued to have good general health perceptions. This is in line with results from the US, showing that for a majority of people, age is not a key explanatory factor for health decline and that many live in good health until older age [7]. As could be expected, a higher number of physician-diagnosed chronic diseases and mental disorders was associated with a higher likelihood of belonging to a trajectory of poorer and declining health as measured by general health perceptions.

One may also consider or question the choice of the socioeconomic position indicator and its effects on the results. Although higher income is associated with more physical activity, in an earlier Finnish birth cohort study, education and occupation emerged as more important determinants of physical activity, and income had no clear effects of its own when it was examined within other socioeconomic groups [49]. However, the participants of that study were in midlife and our participants were significantly older. Our choice of income as an indicator of socioeconomic position can be justified because income reflects resources that are relevant for both of our health-related measures. Accordingly, it has been argued that education, occupational class and income should not be used interchangeably as indicators of a latent social dimension, as they each capture different phenomena, with income potentially providing more general health advantage in the form of material resources, for instance [50]. However, education, occupational class and income correlate, and in addition to covering their specific areas, they also commonly capture socioeconomic position [51]. Moreover, it is not possible to model joint development with any other indicator, as only income clearly changes during follow-up. Changes in education are almost non-existent and occupation is a more relevant indicator during working life, although it may have some effects after retirement [3]. We wanted to specifically focus on whether change in household income was associated with changes in physical activity and general health perceptions. Income typically declines after retirement, and this could contribute to opportunities for leisure-time physical activity or to having the resources to take care of one's health. Furthermore, during working life, all these participants had complimentary access to occupational health services, an advantage which ceases after retirement. Along with declining income, this could explain poorer health, as access to health care, when needed, becomes limited.

Despite examining a broad range of key potential determinants of distinct trajectories, the associations between the used social and health-related factors and trajectory group membership, were mostly modest or non-existent. However, education and BMI were among the clearest determinants of distinct developmental patterns. There is a plausible explanation for

this, as obesity in particular is associated with physical activity [52, 53] and with health [54]. In addition, the educational gradient is clear for both physical activity [55] and self-rated health [15]. In one Finnish study, education and good health were linked to increasing physical activity over a follow-up of 11 years [56]. The link between physical activity and health did not emerge in our models, as physical activity increased in none of the groups, not even among those with stable, good health. Education could be a more consistent predictor of health-related outcomes than income. However, the differences between studies could also be explained by the use of different methodologies and the cohort consisting of older adults. We have also earlier demonstrated with these data that obesity is linked to a later decline in both individual and household income, alongside other indicators of economic disadvantage such as wealth [57]. However, the current study was able to distinguish between different developmental patterns and increase our understanding of the role of education and obesity, which is not homogeneous or determinist for all with a low socioeconomic position. As marital status is a consistent determinant of health [58, 59], also robustly among older adults [60], one might have assumed it to have a more consistent association with trajectory membership. Nonetheless, in line with the known links between marital status, health, and well-being, being single was linked to a lower likelihood of belonging to Group 4, a trajectory characterized by sharply increasing and high income. The participants assigned to this trajectory also reported having the recommended levels of physical activity and good although declining health as measured by general health perceptions. Perhaps somewhat unexpectedly, despite unmarried men in particular being at a higher risk of adverse health outcomes [59], gender did not emerge as a clear determinant of developmental patterns in our outcomes. As our data were from the public sector, which is dominated by women, it is possible that the lack of a stronger association for men is due to lower statistical power. It may also be a unique feature of this cohort. However, many of the associations appeared strong when reported as odds ratios (S3 Table in S1 File), and gender, for example, emerged as a consistent determinant of distinct development, as did marital status and most of the studied social and health-related factors. The odds ratios should be interpreted with caution, particularly the strongest ones, to avoid exaggeration of the associations. Moreover, unlike the AMEs, odds ratios are not comparable across groups [42]. They should also not be misinterpreted as relative risks, which is another reason why we preferred to use AMEs for this study. This also makes the results more concrete and tangible for policy implications. We included a broad range of explanatory factors which could also be each other's determinants and outcomes [61, 62]. Therefore, we examined them separately. Nonetheless, the information on their associations with distinct development may help when, for instance, planning targeted interventions for the identified risk groups.

As this cohort was from the same employer, albeit the largest in Finland, we cannot generalize the findings directly to the private sector, or even to the entire public sector. However, it is difficult to see why the results would notably differ if we had access to similar data from other cities or about persons who have transitioned to statutory retirement from other sectors. Thus, one can only broadly assume that similar patterns might be repeated among public-sector employees and potentially in the private sector. For example, earlier, we have extensively studied sleep, working conditions and sickness absence, as well as the social determinants of sleep, using both these HHS data before the participants retired [63, 64], and nationally representative data on all the working-aged from different sectors [65, 66]. The results were very similar, suggesting that the associations between social and health-related determinants and outcomes in this cohort are comparable with those in nationally representative survey data. Generalizability outside the Finnish context is a further point to consider, because social welfare systems, levels of access to healthcare, and public health measures differ between countries [14, 67]. However, the same points apply as above. With repeated collected data on participants of the

same age, there is no reason why these developmental patterns and their associations with social and health-related factors would only apply in Finland. However, retirement age and retirement systems do differ notably, and this could make a difference due to the age-dependence of the development of general health perceptions [1–4].

## Methodological considerations

Our study was a follow-up survey, based on self-reported data. Thus it cannot be ruled out that some people may have responded negatively to each item, or that their responses to the questions on their leisure-time physical activity, general health perceptions, and household income did not reflect the true situation, but social desirability [68, 69]. However, it is also an advantage that we received information on the different indicators of well-being and the factors associated with them directly from the retirees themselves, on matters that are not available in administrative records. Although objective measures of physical activity could be more accurate, depending on the methods used, it would not be feasible to collect such data from thousands of participants over 22 years. Additionally, no measure of physical activity has proven to be superior [70]. As we were not focusing on the actual levels of physical activity but on the patterns of change and on identifying distinct development, it is likely that the self-reports adequately indicated the different groups of people expected to follow a similar physical activity, general health perceptions, and income trajectory. One could also wonder whether it was difficult for the participants to find suitable equivalents for all their different activities. However, the question included a fairly long introduction and guidelines to assist them (please see S1 File for the full inventory). We acknowledge that we may still have missed some lower intensity activities due to difficulties finding suitable equivalents. As the HHS is a wide-ranging health survey that has been underway since 2000, focusing on socioeconomic inequalities in health, working conditions, health behaviors and different health outcomes, it was lengthy, and we had to compromise on what to include, in order to have valid measures but not an overly long survey. The chosen physical activity measure has also been used in the well-established Finnish Twin Cohort study [27, 71] and other Finnish cohort studies [72]. Gardening and other yard work are generally quite common among midlife and older adults in Finland. However, in this cohort they were likely quite rare, as people seldom have gardens in Helsinki (people mainly live in apartment buildings). Weightlifting or strength training are also generally popular Finland, but these participants were of an age cohort among which weightlifting is not a common hobby. They were most likely to walk, ride a bike, or ski in the winter, or do some other aerobic activities [73, 74]. As they were older adults (the oldest were 82 in 2022), walking was the most common type of activity. In addition, as we were not studying individuals, but groups, missing out some type of activity should not have distorted the findings. The items are likely to provide a hierarchy, and identify people with low, intermediate, or high activity, which in reality may be even higher. Furthermore, our physical activity measure has shown to predict mortality and different health outcomes [75, 76]. Another point to consider are the bidirectional associations that are possible in observational studies. Thus, on the one hand, at any time point, those with more physical activity could report better health, but on the other hand, those with better health are probably able to be more physically active. The advantage of the joint GBTM is that it can identify co-occurring patterns in outcomes over time. In addition, self-reports may provide inaccurate estimates for some covariates. For example, in surveys sleep duration can be under- or over-estimated and body weight is typically under-estimated [77, 78]. However, both are self-reported in reasonable accuracy and have shown association with health measures such as sickness absence in these and nationally representative data [66, 79]. Finally, we acknowledge that some of the covariates were time variant—BMI, for example, may change

over time—and this could have contributed to their associations with trajectory membership. As we had five time points, three outcomes, and many covariates, the models would have been very complex if we had considered both changes in covariates and outcomes at the same time. One key determinant—education—was only elicited in Phase 1 and it does not typically change in later midlife or during the transition to retirement. BMI also emerged as a key determinant, but we have also earlier studied BMI trajectories quite extensively in these data, and found that differences in BMI were clear in young adulthood, but remained very similar from inclusion and throughout follow-up [80]. It would not be possible to have health as an outcome and a covariate simultaneously. For these reasons, we chose to include covariates at the point of inclusion.

The main strength of this study is its large size and long follow-up of up to 22 years. Thus, we were able to follow the same people from 2000 to 2022, and had information on their physical activity, general health perceptions and income from five time points. The respondents to the Phase 1 survey remained in the cohort well, and the response rate was high at each follow-up point. Thus, it is unlikely that attrition notably distorted our findings. Our supplementary analysis on attrition confirmed this (S5 Table in S1 File), as the numbers lost to attrition or due to missing data were low. There were some differences in age, for example, among those lost to attrition, but due to their small numbers, it is unlikely that the results were affected to any great extent. Moreover, our extensive non-response analyses have shown that the data are broadly representative of the target population, although men, younger participants, and those with more sickness absences were somewhat less likely to participate at baseline [22, 81]. These differences were minor, but if we had been able to include the whole target population as respondents, the prevalence of poor health as measured by general health perceptions would probably have been slightly higher. This means that the current results could be somewhat conservative. As the cohort has been shown to adequately represent the target population, small differences in sickness absence are unlikely to have distorted the findings. Another strength of the study is that we repeated all the survey items at each time point, which enabled a true follow-up of the changes and allowed us to model the distinct developmental patterns. Five time points is ideal for trajectory analyses, and enables better detection of different shapes in the trajectories: for instance, the cubic shape is not possible with three time points. Overall, the use of a person-oriented method (joint GBTM) is a strength of the study, as it enabled us to identify the qualitatively distinct development of the outcomes. These latent groups would not be directly observable in the data, for example, with series of repeated measurements of mean values. This means that we could find distinct groups that variable oriented methods might miss, as they assume a homogeneous risk of poor health among all those with low income vs. high income, or similar differences in physical activity or income based on health status or physical activity. It is thus important that prevention measures aim to identify these subgroups of people who have all the risk factors, but also that we improve our understanding of the factors that explain why some groups of people have relatively high scores of general health perceptions despite low income. However, it is also important to highlight that while the method provides easily interpretable visualization of distinct latent groups in the data, instead of using a pre-defined cut-off for all, the groups are only approximations of actual development, and some individuals may have been misclassified. Differences between trajectory models were addressed in our earlier study [38]. Finally, we also had a broad range of key potential social and health-related factors available in these data, which may explain the distinct developments in our outcomes.

As a sensitivity analysis, we conducted all the models using only the four first time points. The numbers were smaller, as more people retired after Phase 4 and could be included in this study. The results would, however, have been very similar regardless of whether we used four

or five phases, and the joint GBTM produced practically identical trajectories. Statistical power is greater with more data, which is reflected as smaller standard deviations in most estimates using five vs. four phases. The average retirement age of the population using five phases was somewhat lower, because the younger generations retired after Phase 4 and were included in this study. The associations between social and health-related factors and trajectory group membership were also similar in both the shorter and longer follow-up. To have up-to-date data and for better GBTM options, we chose to have five time points. The results of the four phases are reported briefly (no further data shown), because one might wonder if other changes had happened during a longer follow-up (unmeasured confounding), or if attrition had been an issue or if with aging, major health-related changes might have distorted the findings. However, none of these speculations are supported by the sensitivity analysis.

## Conclusion

Most participants who transitioned to statutory retirement had relatively good health as measured by general health perceptions, despite changes in leisure-time physical activity and household income. Thus, changes in income did not develop jointly with changes in general health perceptions. Social and health-related factors, particularly education and obesity, showed associations with the trajectory memberships, reflecting known health inequalities. Following the joint development of physical activity, general health perceptions, and household income showed that these do not necessarily develop together, and that some group will always be better off in all indicators, while another will consistently have low physical activity, poor health, and low income. Instead, the development was distinct, and higher socioeconomic position as indicated by income alone did not explain as general health perceptions or stable high physical activity. Physical activity is important, and we identified large groups of retirees who had the recommended level of physical activity and good general health perceptions (without excessive amounts of activity). Thus, after retirement, a moderate, recommended level of activity could be enough to maintain good health. As this study is observational, causal claims are unwarranted. Nonetheless, the model we applied could have identified a decrease in the amount of physical activity prior to change in health in a group, for example, should such a group have existed in the data. This result could be highlighted in health policies to promote the fact that doing the recommended level of physical activity is beneficial in the long run. In other words, our results support and are in line with earlier recommendations that adults should do at least 150 min of moderate-intensity aerobic physical activity, or at least 75 of vigorous-intensity aerobic physical activity, or an equivalent combination of moderate-intensity and vigorous-intensity activity every week for substantial health benefits [11, 82, 83]. Some people also had had relatively high scores in general health perceptions despite a low income over time, and relatively high levels of physical activity. Therefore, health policies and interventions to maintain and improve well-being during aging need to take into account the distinct development in well-being before and after statutory retirement and that an indicator such as income alone is not sufficient when planning targeted measures.

## Supporting information

**S1 File. Supporting S1–S5 Tables and supporting S1–S6 Figs as well as an S1 File including additional methods supplement (survey questions used in the study) are available as a combined separate document.**
(PDF)

## Author Contributions

**Conceptualization:** Tea Lallukka, Petteri Kolmonen, Ossi Rahkonen, Eero Lahelma, Jouni Lahti.

**Formal analysis:** Petteri Kolmonen.

**Funding acquisition:** Tea Lallukka, Ossi Rahkonen.

**Methodology:** Tea Lallukka, Petteri Kolmonen, Jouni Lahti.

**Project administration:** Tea Lallukka.

**Resources:** Tea Lallukka.

**Supervision:** Tea Lallukka, Jouni Lahti.

**Visualization:** Tea Lallukka, Petteri Kolmonen.

**Writing – original draft:** Tea Lallukka.

**Writing – review & editing:** Petteri Kolmonen, Ossi Rahkonen, Eero Lahelma, Jouni Lahti.

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
