## [Decision Letter · Decision Letter 0]

21 Jun 2024

PONE-D-24-14501Joint trajectories of physical activity, health, and income before and after statutory retirement: a 22-year follow-upPLOS ONE

Dear Dr. Lallukka,

Thank you for submitting your manuscript to PLOS ONE. After careful consideration, we feel that it has merit but does not fully meet PLOS ONE’s publication criteria as it currently stands. Therefore, we invite you to submit a revised version of the manuscript that addresses the points raised during the review process.

The manuscript has been evaluated by three reviewers. While two reviewers suggest that the manuscript needs to be revised before it could be accepted for publication, one of the reviewers suggests rejection. Although I agree with the concerns of the latter reviewer, I think there is a possibility that these concerns could be addressed in a major revision.

The main concerns of the reviewers are the lack of clarity and the lack of reproducibility, especially in the modelling of the trajectory. The manuscript also suffers from grammatical errors and awkward phrasing, requiring further language editing.

Based on the review reports and my own assessment as an academic editor, I am pleased to inform you that it is potentially acceptable for publication in PLOS ONE, provided that you make some essential major revisions suggested by our reviewers.

We look forward to receiving your revised manuscript.

Kind regards,

Christoph Strumann

Academic Editor

PLOS ONE

 [This study is supported by the Research Council of Finland (Grant #330527) and OR is supported by the Ministry of Education and Culture, Finland (Grant #OKM/76/626/2023) and by the Juho Vainio Foundation (Grant #202300041)].  

Reviewers' comments:

Reviewer's Responses to Questions

**Comments to the Author**

1. Is the manuscript technically sound, and do the data support the conclusions?

Reviewer #1: Yes

Reviewer #2: Partly

Reviewer #3: Partly

2. Has the statistical analysis been performed appropriately and rigorously? 

Reviewer #1: Yes

Reviewer #2: I Don't Know

Reviewer #3: I Don't Know

3. Have the authors made all data underlying the findings in their manuscript fully available?

Reviewer #1: No

Reviewer #2: No

Reviewer #3: Yes

4. Is the manuscript presented in an intelligible fashion and written in standard English?

Reviewer #1: Yes

Reviewer #2: No

Reviewer #3: No

5. Review Comments to the Author

Reviewer #1: In this survey-based study over a 22-year follow-up among public health workers, the authors investigated the joint trajectories of family income, leisure time, physical activity, and health. They have identified four trajectories and concluded that physical activity and health trajectories are not strictly linked to income trajectories and that even individuals with low-income trajectories can follow high physical activity and good health trajectories, suggesting that income and health might be less strongly correlated in the Finnish setting. The manuscript is well-written, and the method is adequate. However, there are some limitations of the study that the authors should discuss further.

Comments:

# Why was sleeping categorized as such? According to public health recommendations and sleep research, 7-9 hours of sleep is considered healthy and is associated with the lowest mortality.

#The description of the result of the GBTM would fit better in the Results section.

#Can the authors reason why they did not choose the 5 or 6 trajectory solutions, as these still resulted in a larger drop in AIC and BIC and have the lowest log-likelihood function?

#It seems that education is still a strong indicator of both income and health-related trajectories, while income is not correlated that closely with health-related trajectories. This is in line with the results of studies investigating different socioeconomic measures in relation to health outcomes, which I think would be useful to discuss.

# “While high levels of physical activity are consistently linked with better health, this study also suggests that recommended level of physical activity is likely sufficient to maintain health in aging and older adults.” This is an overstatement and suggests causality, which was not possible to investigate with the chosen study design and method.

#One of the largest limitations of this study is the possible selection bias due to selective dropout during the follow-up. Therefore, it would be important to present supportive data that attrition was not selective. I would suggest comparing the characteristics of those who dropped out during the follow-up vs those included in the study to have a clearer picture of this type of bias.

#“Our extensive nonresponse analyses have, nonetheless, shown that the data are broadly representative of the target population, although men, younger participants and those with more sickness absence were somewhat less likely to participate at baseline [21,62].” It would be important to further discuss how the limited participation of those on sickness absence might influence inference to the target population.

#Generalizability to all employers should be discussed, as there might be differences between private and public employment in regard to access to occupational health services, health-related measures at the workplace, such as reimbursement and preventive measures, as well as supported employment for those with health problems, between the sectors.

#Generalizability outside the Finnish context should be discussed, as the social welfare system, access to healthcare, and public health measures differ in other countries/settings.

#Many of the covariates, such as obesity, alcohol use, and sleeping habits, can change over time. How did the author consider changes and possible interaction with the outcome measures?

Conclusion

“This suggests that after retirement, one may not need excessive amounts of physical activity but a moderate, recommended level is likely sufficient to maintain good health.” The study design does not allow conclusions about causality, as the outcomes are measured at the same time. The authors have to be very careful of the interpretation of findings and not translate them into recommendations, given that the study design and method used are not adequate to study causal structures.

Reviewer #2: 1. Outcome of interest for selecting the trajectory models:

Based on the description of the methods, it is unclear to me which outcome was used to select the trajectory groups. As stated in Nagin et. al 2010 referring to both GBTM and GMM “Both methods are designed to identify cluster of individuals, called trajectory groups, who have followed a similar developmental trajectory on an outcome of interest”. Based on Figure 1, there are 3 outcomes of interest: General Health perceptions, Household income, and Leisure-time physical activity. It is unclear how the same 4 trajectory groups were determined for three different outcomes.

2. Language- The paper would be easier to read and understand if the following issues were addressed:

Clarity: There are several very long sentences/ awkwardly worded sentences. Here is one example: “Following the joint development and co-occurrence of physical activity, health and income can help identify if there are groups of people who have little physical activity, poorer health, and low income, or if some groups are continuously better off, or if the development is distinct.”

Grammar: There are multiple grammatical errors throughout the paper. Here is one example: “which enables to approximate”, “The participants further provided an informed consent whether they agreed their survey responses”

I recommend rewording the phrase “somatic and mental diseases”.

3. Description of statutory retirement: While some background is provided in the Methods section, it would be helpful to have more information about statutory retirement in Finland, as many readers may be unfamiliar with this. Specifically, is retirement mandatory at a specific age or does it depend on the type of job? Additionally, to better understand the income of retirees, it would be helpful to have a description of the pension or retirement benefits that retirees in Finland receive.

4. Leisure time physical activity:

a. For study reproducibility and generalizability, please provide the exact survey questions and response options for leisure time physical activity.

b. What was the timeline for the questions about physical activity? Were participants reporting on current activity or recalling previous levels of physical activity?

c. Which specific guidelines for physical activity were used to determine categories for levels of leisure-time physical activity. For example, 2.5 hours of brisk walking per week was categorized as “clearly below recommended leisure-time physical activity” in the discussion without citing a source. 150 minutes of physical activity per week is considered the recommended amount of physical activity for adults, according to some guidelines, including the World Health Organization.

5. Sleep- Is there a reference for choosing 7 hours as the reference category? What happened to participants who slept between 7 and 8 hours? This level of the sleep category is missing from Table 1 and Table 2.

6. Timing of measures

From the abstract and introduction, I understand that there were repeated surveys of participants at each of the 5 phases of the study. However, it is unclear in the methods section, if the “social and health-related covariates” were measured just at baseline, or at each of the 5 study phases. I understand the measurements that were taken at baseline (Table 1.), but were these measurements taken again at other phases of the study? I understand that some of these variables are unlikely to change, however, some variables such as amount of sleep, weight, and comorbidities may change during the study period. Additionally, the duration of time accounted for in the measurements is unclear. For example, what time period do the sleep and physical activity measures cover? Sleep the previous night, week, month, year? This is important in consideration of recall bias, and the stability of these measurements.

7. “All those who responded to at least two if the five follow-up questionnaires were included in this study”.

Given the following recommendation, please provide justification for the inclusion of participants with only two data points.

“In latent class modelling approaches for longitudinal data, at least three measurement time points are required for proper estimations, and four or five measurement time points are preferable in order to estimate more complex models involving trajectories following cubic or quadratic trends”. Trajectory Modelling Techniques Useful to Epidemiological Research: A Comparative Narrative Review of Approaches Clinical Epidemiology Nguena Nguefack, Hermine Lore; Pagé, M Gabrielle; Katz, Joel; Choinière, Manon; Vanasse, Alain; Dorais, Marc; Samb, Oumar Mallé; Lacasse, Anaïs

Vol. Volume 12, pp. 1205–1222, 2020.

8. “Only a fourth had no physically strenuous work history” Please provide a percentage.

9. “Thus, people with obesity had a 23% higher likelihood to belong to the Group 1, as compared to people with normal weight.”- Is this based on weight at baseline or is this time-varying?

10. “While high levels of physical activity are consistently linked with better health, this study also suggests that recommended level of physical activity is likely sufficient to maintain health in aging and older adults.” Please address the potential for a bidirectional relationship between physical activity and health, for example, those with more physical activity may report better health, however, those with better health may be able to be more physically active.

11. “Among the potential factors, education and obesity were key factors explaining the patterns.”

Education likely did not vary over time during the study, given that the participants were retirement age, however, obesity, specifically weight is a time-varying covariate. Were changes in weight over time accounted for? Also, there is a potential for a bidirectional relationship here, “Reporting three or more somatic and mental diseases was associated with a higher likelihood to belong to the Group 1 (26%)”. A higher number of somatic and mental diseases may be associated with an increased likelihood of obesity.

12. “While high levels of physical activity are consistently linked with better health, this study also suggests that recommended level of physical activity is likely sufficient to maintain health in aging and older adults.” Please provide more details about what findings from this study this statement is based on.

13. “Physical activity typically likely declines with aging and is important for health [11] but changes in income did not develop jointly with changes in health in all groups.” This sentence is not very clear, and it would be helpful to separate out the findings for physical aging and income.

14. “We also identified large groups of people – two thirds of the current population - with recommended and high levels of physical activity, continuing to have good health.” Again, please address here the potential for a bidirectional relationship between physical activity and health.

Reviewer #3: Thank you for the opportunity to review the manuscript titled: “Joint trajectories of physical activity, health, and income before and after statutory retirement: a 22-year follow-up”. The study covers an interesting and relevant topic and employs a sophisticated analysis technique. The authors should be lauded for having collected a large volume of data spanning up to 22 years. Nevertheless, I am concerned about some of the analytical and methodological choices, particularly considering this study was not pre-registered. Please find below several suggestions and comments, which I hope will aid you in improving this manuscript:

General comments:

The English language should be improved to ensure that an international audience can clearly understand your text. An example where the language could be improved is (introduction): “However, the above previous variable-oriented studies have not considered latent groups or examined how indicators of well-being develop together, including changes over a major life transition, statutory retirement, following-up same people from their mid- or later careers until older age”. The current phrasing makes comprehension difficult. I suggest you have a colleague who is proficient in English and familiar with the subject matter review your manuscript, or contact a professional editing service.

Could you comment on whether it is problematic that the data collection for phase one spanned three years (instead of one year for subsequent phases), meaning that the time difference between phase 1 and 2 ranges from five to seven years?

It would be beneficial for the reader to know how you defined retirement. Perhaps also mention whether there is a mandatory retirement age in Finland.

I am concerned about your decision to include all participants who completed two surveys. This means that some participants are followed for 5 years and others for 22 years. Greater changes are likely to occur if participants are followed for a longer period of time and older participants are more likely to drop out due to death or other factors during that time. Could you perhaps explain how you addressed this difference? Further, and this is a minor point, I would argue that Phase 1 is the baseline survey, which means that the four subsequent surveys, or phases, are the actual follow-ups, meaning there are only four follow up phases.

Do you have a code/number available for the ethics approval? I think it would be helpful for this to be mentioned here.

I think the outcomes section could be greatly improved by providing more detail. Ideally, provide the exact wording of the assessment, either in the text or an appendix, so the reader knows exactly what you assessed and over what period.

It seems odd you would assess leisure time physical activity by solely asking about walking, brisk walking, jogging and running or equivalents. Leisure time physical activity is a broad category and may comprise gardening as well as heavy weight lifting, for which it may be difficult to find these equivalents. Overall, this would not be an adequate representation of leisure time physical activity.

I think the “Health” section would be more appropriately labelled as self-perceived health. I think you cannot use this synonymously with health. This has implications for the remainder of the manuscript, especially the discussion section.

Please, could you provide more information on the education categories and how these were derived?

You state: “We also adjusted for physically strenuous work before retirement transition and age during the first study wave.” How was this done? How did you define physically strenuous work?

Why was a cut-off of 7 hours per day chosen to assess sleep?

Is the BMI measurement adjusted to include participants who can be categorised as underweight?

Could you explain why never or ex-smokers were conflated into one category?

Could you elaborate why you chose binge drinking behaviour instead of alcohol consumption at various levels? There appear to be few participants actually engaging in binge drinking behaviour, which makes me contemplate the usefulness of this.

Regarding the disease count, what was the time span that was assessed? Was it the five years prior?

Overall, the outcomes section could greatly benefit from more detail pertaining to how details were assessed and over what time span.

Were there any missing data and if so, how were these handled?

It is not clear to me how you determined the change from employment to retirement. Presumably this varied between individuals. Did you follow individuals as they aged, i.e., as their age increased irrespective of phase, or did you use subsequent survey participations as an indicator of time passed. The latter may be problematic, given the range of ages represented. This question may be rooted in a misunderstanding of the analytical methods employed, but nevertheless, it is worth clarifying this issue for the reader.

Table 1 indicates everyone took part in Phase 1, which is seems at odds with your stipulation of participants needing to have participated in two follow up surveys, which may exclude phase 1. Please can you clarify this? Were any participants added at subsequent phases?

Please include Figure 1.

Can you comment on why household income is not represented in tables 1-2?

You discuss increases and decreases in various measures. Could you comment on the size and magnitude of these?

What was the average number of years that participants were followed up for?

6. PLOS authors have the option to publish the peer review history of their article (what does this mean?). If published, this will include your full peer review and any attached files.

Reviewer #1: No

Reviewer #2: No

Reviewer #3: No

---

## [Author Response · Author response to Decision Letter 0]

26 Sep 2024

Christoph Strumann

Academic Editor

PLOS ONE

RESPONSE: We ensure that the manuscript meets the PLOS ONE’S style requirements, as requested.

 [This study is supported by the Research Council of Finland (Grant #330527) and OR is supported by the Ministry of Education and Culture, Finland (Grant #OKM/76/626/2023) and by the Juho Vainio Foundation (Grant #202300041)]. 

RESPONSE: Thank you. We confirm that the statement is correct. The funders had no role in this study, and we have included the statement, as requested.

RESPONSE: These data are highly sensitive, and we are not permitted by law to share these data (GDPR). These data can only be used by the research group, on a secure server. Our data protection document available our webpage includes all the details: https://www.helsinki.fi/en/researchgroups/helsinki-health-study/data-protection-statement.

Reviewers' comments:

Reviewer #1

In this survey-based study over a 22-year follow-up among public health workers, the authors investigated the joint trajectories of family income, leisure time, physical activity, and health. They have identified four trajectories and concluded that physical activity and health trajectories are not strictly linked to income trajectories and that even individuals with low-income trajectories can follow high physical activity and good health trajectories, suggesting that income and health might be less strongly correlated in the Finnish setting. The manuscript is well-written, and the method is adequate. However, there are some limitations of the study that the authors should discuss further.

Comments:

# Why was sleeping categorized as such? According to public health recommendations and sleep research, 7-9 hours of sleep is considered healthy and is associated with the lowest mortality.

RESPONSE: We appreciate this comment. Our sleep measure has a minor role in this study. It is but a covariate and it is rather crude. Unfortunately, sleep duration is asked to be reported only in full hours. For meaningful distribution, and to be able to distinguish between short, average, and long sleepers, we initially chose those responding 7 hours as a reference group (average sleep), while those sleeping less (6 hours or 5 hours or less) as short sleepers and those sleeping more than 7 hours (8 hours, 9 hours, or 10 hours or more) as long sleepers. A vast majority slept 7 or 8 hours (72.4%). For older adults, recommendations is 7–8 hours (https://www.sciencedirect.com/science/article/abs/pii/S2352721815000157 & http://dx.doi.org/10.1016/j.sleh.2015.10.004) and for young adults 7–9 hours. Additionally, if we used 7–9 hours as a reference, almost all participants would be placed in this group (75.9%). Almost nobody reported sleeping 10 hours or more (0.2%, 10 participants), and thus we would not be able to study long sleep at all. In sleep research, 7 and 8 hours are often combined, or even 6–8 hours to reflect average sleep, but as this cohort was employed at inclusion, it was extremely rare to report very short or long sleep. To have meaningful category sizes, and to somehow (with limitations), distinguish between short, average, and long sleepers, we thus chose this classification. If this was a sleep study, we fully agree we would need more detailed measures, or if we were interested in individual level sleep. However, for a covariate, and epidemiological purposes, we trust this variable works as a proxy for sleep quantity. We have amended the revised Methods to be clearer in describing these measures, and all our covariates and outcomes (following other comments). We now use 7–8 hours as a reference, to better meet with the recommendations.

We have clarified the methods and updated the classification following this comment (please see page 9).

#The description of the result of the GBTM would fit better in the Results section.

RESPONSE: Thank you for this comment. We are a bit unsure what it means, as all the results of the model are in the Results section. The statistical analysis and model selection criteria are described in the Methods section. It is how these model selection criteria are usually reported in previous literature, and it would feel somewhat strange to report statistical parts as results. We have thus retained the original place, as most of the description is technical and justifies the selected model. Description for other models tested would be confusing as results and it is difficult to separate statistical parts for the selected and other models.

#Can the authors reason why they did not choose the 5 or 6 trajectory solutions, as these still resulted in a larger drop in AIC and BIC and have the lowest log-likelihood function?

RESPONSE: It is true that often the models with more classes may appear superior if one only examines some statistical criteria such as AIC and BIC. Statistical criteria often tend to suggest more complex models with a larger number of classes. However, in trajectory analyses, there are many criteria guiding the selection of the best modes. A key one is related to class sizes: they have to be meaningful, as if the classes are really small, there is little population level relevance, and it is difficult to conduct any further analyses such as study determinants of trajectory membership. Additionally, the classes have to visibly distinct, which sometimes is not that obvious, even if some of the statistical criteria suggest those more complex solutions. Finally, we have to consider average posterior probabilities (APPs) and misclassification. Thus, there must be a meaningful story, and large enough classes (typically at least 100 individuals, or 5% or 10% class). AIC and BIC alone are unlikely to be sufficient in selecting the best model.

In our study, statistically, we might have chosen 5 or 6 class model even, but these did not add new information and all the groups were not as clearly distinct. In other words, when the number of classes increases, some classes are no longer clearly distinct as in our selected 4 class model. Some categories are also small, and it is difficult to examine associations, or provide population level messages.

We regret that our initial appendices did not clearly illustrate all these criteria for more complex models, or provided sufficient explanation for interpretation of the patterns. We have clarified the Methods section to better justify the 4-class solution and have included a more detailed Supplementary file (Fig S3) and examples of 5 and 6 class models in our supplementary material.

#It seems that education is still a strong indicator of both income and health-related trajectories, while income is not correlated that closely with health-related trajectories. This is in line with the results of studies investigating different socioeconomic measures in relation to health outcomes, which I think would be useful to discuss.

RESPONSE: We agree that education and income are not interchangeable and have different associations with health. It is true that education correlated with income and has clear and consistent associations with health outcomes, as mentioned in the manuscript. We have amended the revised manuscript to better highlight this. It is of note though that income is an outcome in the model while education is a from early adulthood, and asked to be reported at Phase 1 only, and it is only used in the multinomial models. As can be seen in the Figure, income changes over time and there are distinct groups among the participants. Education was included as a covariate in the model where income is one of the outcomes. Thus, one has to consider that education (covariate) and income (outcome) have very different roles in this study, and in the model with education, there is always income as an outcome, with health-related outcomes.

# “While high levels of physical activity are consistently linked with better health, this study also suggests that recommended level of physical activity is likely sufficient to maintain health in aging and older adults.” This is an overstatement and suggests causality, which was not possible to investigate with the chosen study design and method.

RESPONSE: We fully agree we cannot suggest causality in this design. We have revised the section to be more cautious and in line with the study design, as suggested. Our intention is not to suggest causality. Joint group-based trajectory model is descriptive and only approximates true developmental patterns. It is, however, able to identify latent groups in the data and show if the chosen outcomes jointly co-occur. For example, the model could find people with declining health after they report a change in income or physical activity, if there was such a group in the data.

#One of the largest limitations of this study is the possible selection bias due to selective dropout during the follow-up. Therefore, it would be important to present supportive data that attrition was not selective. I would suggest comparing the characteristics of those who dropped out during the follow-up vs those included in the study to have a clearer picture of this type of bias.

RESPONSE: Thank you for this suggestion. This cohort actually has exceptionally high response rate throughout the follow-up. Thus, attrition is not a major issue, and our model can also use the data from the timepoints available. Different follow-up time does not affect this model. This choice avoids redundant selection as each participant could contribute to the trajectories even if they did not have all 5 time points (a vast majority though did have 4–5 time points and only 0.1% had 2 time points, they are now excluded). However, we agree that attrition can still be selective, and one could assume that those with the poorest health, low physical activity and low income are over-represented among those lost to follow-up. If they still contributed to at least 3 phases, their responses could be used. We have studied loss to follow-up by the key variables of interest (gender, age, marital status, education, physically strenuous work, smoking, binge drinking, sleep duration, obesity, physician diagnosed diseases, income, leisure-time physical activity, and general health) and the results are included as a new Supplement Table S5 and briefly covered in the revised Discussion (please see page 23). These covariates were taken from Phase 1 and then we examined loss to follow-up as an outcome (lost to follow-up =1, included in this study =0). Those who were working or on disability pension were excluded from this study and were not among those lost to follow-up but outside the study scope.

#“Our extensive nonresponse analyses have, nonetheless, shown that the data are broadly representative of the target population, although men, younger participants and those with more sickness absence were somewhat less likely to participate at baseline [21,62].” It would be important to further discuss how the limited participation of those on sickness absence might influence inference to the target population.

RESPONSE: We have included more details on this. The differences were minor as reported in the published studies. However, it is likely that if we had all those in the target population as respondents, health would be slightly worse. Thus, the results could be somewhat conservative. To highlight, the cohort has been shown to represent the target population quite well, in several studies, so these differences are unlikely to distort the findings to any larger extent.

#Generalizability to all employers should be discussed, as there might be differences between private and public employment in regard to access to occupational health services, health-related measures at the workplace, such as reimbursement and preventive measures, as well as supported employment for those with health problems, between the sectors.

RESPONSE: As this cohort is from one employer, albeit the largest one in Finland, we cannot generalize the findings directly to private sector, or even to the entire public sector. However, it is difficult to see why the results would notably differ in other occupational cohorts in Finland if they were to examine developmental patterns in these outcomes among statutory retirees. Individual levels of physical activity or income are likely different, but as this is an epidemiological study, reporting findings at a group level, one could assume that broadly similar patterns might be repeated in public sector employees and potentially in private sector. For example, we have earlier extensively studied sleep, working conditions and sickness absence as well as social determinants of sleep, using both these HHS data before the participants retired, and also nationally representative large data (representative of all working-aged from different sectors, please see links to the studies here below). The results have been highly similar, suggesting that there are very similar associations between social and health-related determinants and outcomes, both in this cohort and using a nationally representative survey data. We have amended to revised Discussion to comment on these points (please see page 20 and 21). However, we prefer to retain these speculations minimal in the text, as we cannot study these current associations in a nationally representative cohort. These analyses can only be done with repeatedly collected data which are rarely available.

References: https://bmcpublichealth.biomedcentral.com/articles/10.1186/1471-2458-12-565 & https://academic.oup.com/sleep/article/37/9/1413/2416837?login=false

#Generalizability outside the Finnish context should be discussed, as the social welfare system, access to healthcar

---

## [Decision Letter · Decision Letter 1]

6 Nov 2024

PONE-D-24-14501R1Joint trajectories of physical activity, health, and income before and after statutory retirement: a 22-year follow-upPLOS ONE

Dear Dr. Lallukka,

Thank you for submitting your manuscript to PLOS ONE. After careful consideration, we feel that it has merit but does not fully meet PLOS ONE’s publication criteria as it currently stands. Therefore, we invite you to submit a revised version of the manuscript that addresses the points raised during the review process.

The manuscript has been evaluated again by the same three reviewers. While two reviewers accept the manuscript in its current version, Reviewer3 has some minor comments that need to be addressed before it could be accepted for publication.

We look forward to receiving your revised manuscript.

Kind regards,

Christoph Strumann

Academic Editor

PLOS ONE

Journal Requirements:

Reviewers' comments:

Reviewer's Responses to Questions

**Comments to the Author**

1. If the authors have adequately addressed your comments raised in a previous round of review and you feel that this manuscript is now acceptable for publication, you may indicate that here to bypass the “Comments to the Author” section, enter your conflict of interest statement in the “Confidential to Editor” section, and submit your "Accept" recommendation.

Reviewer #1: All comments have been addressed

Reviewer #2: All comments have been addressed

Reviewer #3: (No Response)

2. Is the manuscript technically sound, and do the data support the conclusions?

Reviewer #1: Yes

Reviewer #2: (No Response)

Reviewer #3: Partly

3. Has the statistical analysis been performed appropriately and rigorously? 

Reviewer #1: Yes

Reviewer #2: (No Response)

Reviewer #3: Yes

4. Have the authors made all data underlying the findings in their manuscript fully available?

Reviewer #1: No

Reviewer #2: (No Response)

Reviewer #3: No

5. Is the manuscript presented in an intelligible fashion and written in standard English?

Reviewer #1: Yes

Reviewer #2: (No Response)

Reviewer #3: Yes

6. Review Comments to the Author

Reviewer #1: (No Response)

Reviewer #2: (No Response)

Reviewer #3: Thank you for the opportunity to re-review this interesting manuscript on the joint trajectories of physical activity, health, and income before and after statutory retirement.

The English language has been widely improved throughout the manuscript, as has the manuscript overall. Nevertheless, some issues remain that need to be addressed.

I think one of the main issues with this study is that the variables you set out to examine in the introduction do not adequately map on to the actual variables in your data set without limitations. Given the nature of the data I can follow the authors’ arguments for operationalising the variables and data in the way that they did. Nevertheless, these compromises may merely mask some of the underlying more fundamental issues with some of the variables. In the case of leisure time physical activity, for example, the authors acknowledge that they may miss certain activities due to the way this was assessed, but suggest that due to this being a co-variate, it does not invite the same level of scrutiny. I would argue that being a covariate does not mean that a different level of scrutiny should be applied, especially given the fact that PA does form part of the groups that were identified and is mentioned in the title as a main factor. This also applies to BMI and especially the sleep assessment. While the authors do acknowledge the limitations for PA assessment in the discussion, this should be made similarly clear for BMI and sleep assessment.

On a related note, not referring to self-reported health as such is misleading. This is especially true for health as there are vast differences between subjective and objective reports. I think this especially applies to your discussion, where you use the term “health”, without qualifying this to be self-reported. I understand that all of the measures in this study are technically self-reported. Therefore, perhaps a better term would be self-rated health or similar (to distinguish from other variables) that encapsulates the fact that participants were asked to estimate their own health and disease susceptibility, rather than listing conditions or using a composite measure to indicate health.

The self-report nature of retirement is another significant issue and the implications associated with this should be made clearer in the limitations section of the manuscript. I appreciate the difficulty in defining retirement, especially considering changes in legislation, but especially given the complexity of the system there is great potential for confusion, even among participants themselves. It is, for example, not clear whether participants would consider themselves retired when in they were “semi-retired”, perhaps still working, but only part time and in a different job. These limitations need to be discussed and the implications evaluated.

Please refer directly to the supplementary materials when introducing education categories. These may be particularly confusing to non-Finish readers.

7. PLOS authors have the option to publish the peer review history of their article (what does this mean?). If published, this will include your full peer review and any attached files.

Reviewer #1: No

Reviewer #2: No

Reviewer #3: No

---

## [Author Response · Author response to Decision Letter 1]

16 Dec 2024

Reviewer #3

Thank you for the opportunity to re-review this interesting manuscript on the joint trajectories of physical activity, health, and income before and after statutory retirement.

The English language has been widely improved throughout the manuscript, as has the manuscript overall. Nevertheless, some issues remain that need to be addressed.

I think one of the main issues with this study is that the variables you set out to examine in the introduction do not adequately map on to the actual variables in your data set without limitations. Given the nature of the data I can follow the authors’ arguments for operationalising the variables and data in the way that they did. Nevertheless, these compromises may merely mask some of the underlying more fundamental issues with some of the variables. In the case of leisure time physical activity, for example, the authors acknowledge that they may miss certain activities due to the way this was assessed, but suggest that due to this being a co-variate, it does not invite the same level of scrutiny. I would argue that being a covariate does not mean that a different level of scrutiny should be applied, especially given the fact that PA does form part of the groups that were identified and is mentioned in the title as a main factor. This also applies to BMI and especially the sleep assessment. While the authors do acknowledge the limitations for PA assessment in the discussion, this should be made similarly clear for BMI and sleep assessment.

Response: We agree that self-reported questionnaire data always have certain limitations and measured data could provide more detailed information. However, it is quite difficult to obtain measured data for thousands of participants across long time periods. PA is one of our outcomes and of more importance than covariates, as we model its development over time. Therefore, we have aimed to acknowledge the limitations of the measures. Yet no measure is superior (van Poppel et al. 2010). BMI and sleep were used only in the multinomial model to study their associations with trajectory membership, however, we agree it is important to be open about their limitations for correct conclusions regarding their importance to the developmental patterns. For instance, our Phase 1 survey data were compared to measured health check-up data and self-reported and measured BMI showed relatively high correlation and similar associations with subsequent sickness absence (Korpela et al. 2013). Previous research has shown that self-reported sleep duration is a crude measure and people tend to over-estimate their sleep duration showing moderate correlation with actigraphy measures for instance (Lauderdale et al. 2008). In turn, people with insomnia may under-estimate their sleep (Sivertsen et al., 2009). Yet self-reported sleep duration likely provides reasonable estimates for distinguishing different groups i.e. short, average, and long sleepers, and, for instance, sleep duration has shown association with sickness absence using nationally representative data (Lallukka et al. 2014). We have extensively discussed the limitations regarding our self-reported physical activity measure, and improved discussion on our covariates i.e. BMI and sleep as requested.

References:

Korpela K, Roos E, Lallukka T, Rahkonen O, Lahelma E, Laaksonen M. Different measures of body weight as predictors of sickness absence. Scand J Public Health. 2013 Feb;41(1):25-31.

Lauderdale DS, Knutson KL, Yan LL, Liu K, Rathouz PJ. Self-reported and measured sleep duration: how similar are they? Epidemiology. 2008 Nov;19(6):838-45.

Lallukka T, Kaikkonen R, Härkänen T, Kronholm E, Partonen T, Rahkonen O, Koskinen S. Sleep and sickness absence: a nationally representative register-based follow-up study. Sleep. 2014 Sep 1;37(9):1413-25.

van Poppel MN, Chinapaw MJ, Mokkink LB, van Mechelen W, Terwee CB. Physical activity questionnaires for adults: a systematic review of measurement properties. Sport Med. 2010;40: 565–600.

Sivertsen B, Øverland S, Pallesen S, et al. Insomnia and long sleep duration are risk factors for later work disability. The Hordaland Health Study. J Sleep Res 2009;18:122-8.

On a related note, not referring to self-reported health as such is misleading. This is especially true for health as there are vast differences between subjective and objective reports. I think this especially applies to your discussion, where you use the term “health”, without qualifying this to be self-reported. I understand that all of the measures in this study are technically self-reported. Therefore, perhaps a better term would be self-rated health or similar (to distinguish from other variables) that encapsulates the fact that participants were asked to estimate their own health and disease susceptibility, rather than listing conditions or using a composite measure to indicate health.

Response: We have now used general health perceptions throughout the manuscript to avoid confusion. The RAND-36 has 8 dimensions that assess different aspects of physical and mental health: physical functioning, role limitations due to physical health problems, role limitations due to personal or emotional problems, emotional well-being, social functioning, energy/fatigue, and general health perceptions (Hays 1993 & Hays 2001). The term suggested by the developers of the RAND-36 is preferred to self-rated health, as self-rated health is commonly used to refer to a single item measure derived from the same RAND-36 (the first item of the measure), and referenced in the paper (references 31, 32 and 33). This could be confusing and make comparisons between studies difficult. We trust that adding the full-term general health perceptions clarifies the measure is self-reported, as it refers to perceptions. Moreover, the Methods are specific and explicit about the nature of the data and that all items are from surveys. We also explicitly state in the revised Methods section that the subdimension is self-reported. Using the term as measured by general health perceptions makes the text longer and a bit harder to read, as the term cannot be as conveniently used together with good or poor, as before (good health vs. poor health), however, we hope the term is now clear and less misleading.

References:

Hays RD, Sherbourne CD, Mazel RM. The rand 36-item health survey 1.0. Health Econ. 1993;2(3):217-27.

Hays RD, Morales LS. The RAND-36 measure of health-related quality of life. Ann Med. 2001;33(5):350-7.

The self-report nature of retirement is another significant issue and the implications associated with this should be made clearer in the limitations section of the manuscript. I appreciate the difficulty in defining retirement, especially considering changes in legislation, but especially given the complexity of the system there is great potential for confusion, even among participants themselves. It is, for example, not clear whether participants would consider themselves retired when in they were “semi-retired”, perhaps still working, but only part time and in a different job. These limitations need to be discussed and the implications evaluated.

Response: We have now clarified that only full-time statutory retirees were included. In Finland, full-time old-age retirement is probably self-reported with high accuracy due to the structured pension systems, and clear retirement age thresholds. However, we acknowledge that some uncertainty exists in situations such as where people are retired due to old-age and still continue to work part-time. As they still continue to have work-related exposures and may for example commute to work, they are not directly comparable to full time retired participants.

Please refer directly to the supplementary materials when introducing education categories. These may be particularly confusing to non-Finish readers.

Response: We have now clarified this in the text and refer to the supplement, as requested (survey items).

---

## [Editor Report · Decision Letter 2]

20 Dec 2024

Joint trajectories of physical activity, health, and income before and after statutory retirement: a 22-year follow-up

PONE-D-24-14501R2

Dear Dr. Lallukka,

We’re pleased to inform you that your manuscript has been judged scientifically suitable for publication and will be formally accepted for publication once it meets all outstanding technical requirements.

Kind regards,

Christoph Strumann

Academic Editor

PLOS ONE

---

## [Editor Report · Acceptance letter]

29 Dec 2024

PONE-D-24-14501R2 

PLOS ONE

Dear Dr. Lallukka, 

I'm pleased to inform you that your manuscript has been deemed suitable for publication in PLOS ONE. Congratulations! Your manuscript is now being handed over to our production team.

Kind regards, 

on behalf of

Dr. Christoph Strumann 

Academic Editor

PLOS ONE